# Using CMOS Image Sensors to Determine the Intensity of Electrical Discharges for Aircraft Applications

**Jordi-Roger Riba** [1,*] **, Pau Bas-Calopa** [1] **, Yassin Aziz Qolla** [2] **, Marc Pourraz** [2] **and Burak Ozsahin** [3]

1   Electrical Engineering Department, Universitat Politècnica de Catalunya, 08222 Terrassa, Spain
2   Graduate School of Engineering, Polytech Nantes, Rue Christian Pauc, 44300 Nantes, France
3   Faculty of Engineering, Gebze Technical University, 41400 Gebze, Turkey
*   Correspondence: jordi.riba-ruiz@upc.edu; Tel.: +34-937-398-365

**Abstract:** The development of more electric aircrafts (MEA) and all electric aircrafts (AEA) inevitably implies an increase in electric power and a consequent increase in distribution voltage levels. Increased operating voltages coupled with low pressure in some areas of the aircraft greatly increase the chances of premature insulation failure. Insulation failure manifests itself as surface discharges, arc tracking, arcing, and disruptive or breakdown discharges, in order of increasing severity. Unfortunately, on-board electrical protections cannot detect discharges at an early stage, so other strategies must be explored. In their early stage, insulation faults manifest as surface and corona discharges. They generate optical radiation, mainly in the near-ultraviolet (UV) and visible spectral regions. This paper focuses on a method to detect the discharges, locate the discharge sites, and determine their intensity to facilitate predictive maintenance tasks. It is shown that by using small size and low-cost image sensors, it is possible to detect, locate, and quantify the intensity of the discharges. This paper also proposes and evaluates the behavior of a discharge severity indicator, which is based on determining the intensity of digital images of the discharges, so it can be useful to apply predictive maintenance tasks. The behavior and accuracy of this indicator has been tested in the laboratory using a low-pressure chamber operating in the pressure range of 10–100 kPa, which is characteristic of aircraft applications, analyzing a needle-plane air gap geometry and using an image sensor. The proposed method can be extended to other applications where electrical discharges are an issue.

**Keywords:** electric discharge; corona discharge; image sensor; state of health; fault diagnosis

## 1. Introduction

The strict administrative regulations related to the reduction of $CO_2$ equivalent emissions are pushing the aeronautical sector to develop new generations of aircraft with less environmental impact, greater efficiency, lower fuel consumption, or easier maintenance than their predecessors. In this sense, the aircraft industry is developing alternatives, such as the more electric aircraft (MEA) and all electric aircraft (AEA), which, due to demanding requirements in terms of electric power, require operation at higher distribution voltage levels [1], mainly to limit the weight of cabling systems. It is well known that increased voltage levels produce higher electric stress in electric and electronic components and in insulation materials. According to various sources, future aircraft models could operate at higher voltages, in the range from 1 kV [2] up to 4.5 kV [3], which, added to the extreme environmental conditions and high compactness ratios typical of aircraft applications, generate severe electric stress on insulation systems. The higher electric stress increases the intensity and the risk of occurrence of different types of discharges [4], such as surface discharges, arc tracking, arcing, and finally disruptive discharges, which produce different effects, including acoustic and electromagnetic emissions, ultraviolet (UV) and visible light emissions, induction of current pulses in electric circuits, different chemical reactions, or heating, among others. The intensity of the corona spectral bands is known to increase

with supply voltage [5]. The persistent effect of the discharges in insulation materials induces chemical changes resulting from the flow of electrons at the discharge locations [6], increasing their temperature, thus progressively degrading insulation systems, which in turn favors discharge occurrence and eventually leads to major failures [7].

Partial discharges (PDs) occur between two electrodes insulated by a solid, liquid, or gaseous insulation when the local electric field strength exceeds the dielectric strength of the insulation material. PDs do not entirely bridge the insulation path between the electrodes [8]. In the case of gaseous insulation, they are known as corona, which are localized and luminous discharges that occur near an electrode, triggered by a highly inhomogeneous electric field [9] that exceeds a critical value [10,11]. Contrarily, disruptive discharges completely bridge the insulation path between two electrodes [8]. In non-uniform air gaps, corona appears at a lower voltage than the one required to produce a disruptive discharge [12]. In the case of uniform electric fields in air, disruptive discharges appear without previous corona activity.

Although in the short term persistent PDs do not usually pose a serious threat to polymeric insulation in cabling systems, in the long term, due to continuous chemical changes, because of the flow and bombardment of released electrons [6], they can cause serious damage. The chemical changes eventually create a conductive track in the polymeric insulation, which favors the flow of an electric current, thus raising the temperature in the track and causing further damage in the insulation. This sustained situation weakens the insulation and promotes arcing activity [13], which appears in the insulation between wires due to the formation of a carbonized track. Further discharges spread existing tracking paths, generating the arc tracking phenomenon [7], thus extending the damage and eventually causing a fire hazard [14].

Cruising altitudes of commercial jetliners are in the range of 10–13 km, while some other aircrafts have altitude ceilings greater than 15 km, so that operating pressure can be reduced to 10% of the sea level value. It is well known that in atmospheric air, the minimum voltage level at which discharges appear decreases with pressure [15,16], so electric and electronic circuits placed in unpressurized aircraft areas are more susceptible to incidence of electrical discharges [17,18] compared to sea level conditions [19]. It is known that surface degradation of polymeric samples increases as pressure decreases because discharge activity accelerates at lower pressure [20].

For safety and maintenance reasons, and due to strict regulations, the risk of electrical discharges in airplanes must be minimized. Commercial widebody jetliners have complex harnesses, hundreds of kilometers of wires, and thousands of wiring components, such as connectors, clamps, or terminations. Most of these components are hidden inside conduits, ducts, or troughs [21], so maintenance and failure detection tasks become difficult to apply. Due to their low intensity, electrical discharges go unnoticed by the on-board electrical protections, especially at an early phase. However, their persistent effect degrades insulation materials [22], eventually leading to major failures [23]. Therefore, combined actions, such as an improved design and a plan to monitor the discharges, are required to minimize the severity and occurrence of failures. Monitored data enables application of predictive maintenance actions [18] while increasing aircraft safety and allowing application of remedial strategies before critical faults develop [24].

Wiring systems are critical elements for the occurrence of discharges [7], which are often placed in environments with high levels of acoustic and electromagnetic noise, where conventional acoustic or electromagnetic systems to detect partial discharges would fail. It is in these applications where optical sensors are attractive due to their high immunity to electromagnetic and acoustic noise. Optical sensors detect the light produced during the discharge process [25] due to the different transitions of excitation, ionization, and recombination of electrons induced by the discharge [26]. In the case of using optical image sensors, it is also possible to locate and quantify the severity of the discharges, as proposed in this paper, since it is essential for the reliable, safe, and stable operation of aircraft power systems [27]. This approach makes it possible to study the temporal evolution of discharges,

thus opening the possibility of developing state of health (SoH) and remaining useful life (RUL) methods, which are the basis for applying predictive maintenance plans. This field requires more research and experimental data because there is very little experimental work using image sensors for SoH and RUL applications in insulation systems. By analyzing the temporal evolution of the discharges, SoH and RUL tools could provide key information so that remedial actions can be taken before major failures are triggered.

Image sensors are increasingly used to detect corona discharges. In [28], the damage caused by corona discharges to polymeric insulating samples was located and analyzed by means of digital photographs, while in [29,30], the damage caused by mist and acidic fog to polymeric samples exposed to corona discharges was evaluated by means of digital images. In [31], it was revealed that the intensity of the discharges is correlated with the number of photons generated. Using a different approach, in [27], it was shown that under direct current (dc) supply, the electric power dissipated by the PDs is related to the energy of the digital images of the discharges. In [32], the luminosity content of digital images of alternating current (ac) corona discharges was studied and correlated with the measured electric power due to discharge activity. However, [32] lacks a precise analysis, which is required due to the complexity of the electric signals of the discharges, especially due to their low intensity, short duration, and high-frequency content.

This paper contributes in different ways to the area of electrical discharges under ac supply at low pressure, in the range of 10–100 kPa, which is representative of aeronautical applications. First, the magnitude of the fault is measured by applying two approaches, and it is proved that they are directly related. On the one hand, the electric energy dissipated by electric discharges is calculated by means of electric measurements and suitable data processing. On the other hand, the photographs of the discharges are acquired with a digital camera, and their intensity is determined after suitable processing of the digital images. Next, it is shown that the electric energy dissipated by the discharges and the intensity of the digital images are directly related. Finally, a simple and quick to calculate discharge severity indicator is defined based on the intensity of the discharges detected by the image sensor, since it is a key element to quantify the magnitude of the fault, a novelty in this field, which can be very useful for applying predictive maintenance strategies. Although this paper focuses on the aeronautical sector, the results presented may also be interesting in other areas where electrical discharges are a problem, such as plasma surface treatments or to monitor power lines and cables installed at high altitude.

The rest of the article is organized as follows. Section 2 details the signal processing methods applied to determine the electric energy dissipated by electrical discharges and the intensity of discharges in digital images. Section 3 introduces the discharge severity indicator. Section 4 details the experimental setup required to perform the experiments. Section 5 presents the experimental data obtained in a low-pressure chamber and discusses the results obtained. Finally, Section 6 concludes the document.

## 2. Applied Signal Processing Techniques

This section describes the methods proposed in this paper to determine the electric energy dissipated by the discharges and the intensity of the discharges from the digital images provided by a complementary metal oxide semiconductor (CMOS) image sensor. In a practical application, it is not necessary to determine the electric energy associated with the discharges, but here it is done to verify that it is directly related to the intensity of the digital images of the discharges.

### 2.1. Electric Energy Dissipated by the Electrical Discharges

The average or active power of a circuit is defined as the average value of the instantaneous power, which in discrete form can be expressed as [32]

$$P = \frac{1}{tot} \sum_{k=1}^{tot} v(k\Delta t) i(k\Delta t) \tag{1}$$

where $v(k)$ and $i(k)$ are, respectively, the instantaneous values of the voltage and the current in the circuit, $k\Delta t$ is the instant of time in which the instantaneous power is calculated, and $\Delta t$ is the time step of the experimental acquisitions of voltage and current.

Although (1) is easy to apply, some considerations are required due to the high-frequency components and the short duration of the discharges. First, the discharges appear in different points in the waveform, initially appearing during the negative semi period of the voltage, and as the voltage increases, they also appear during the positive semi period. It means that each discharge has its own voltage, which is not constant during the duration of the discharge. Second, due to their low level, it is easier to distinguish the discharges in the current waveform than in the voltage waveform, as seen in Figure 1.

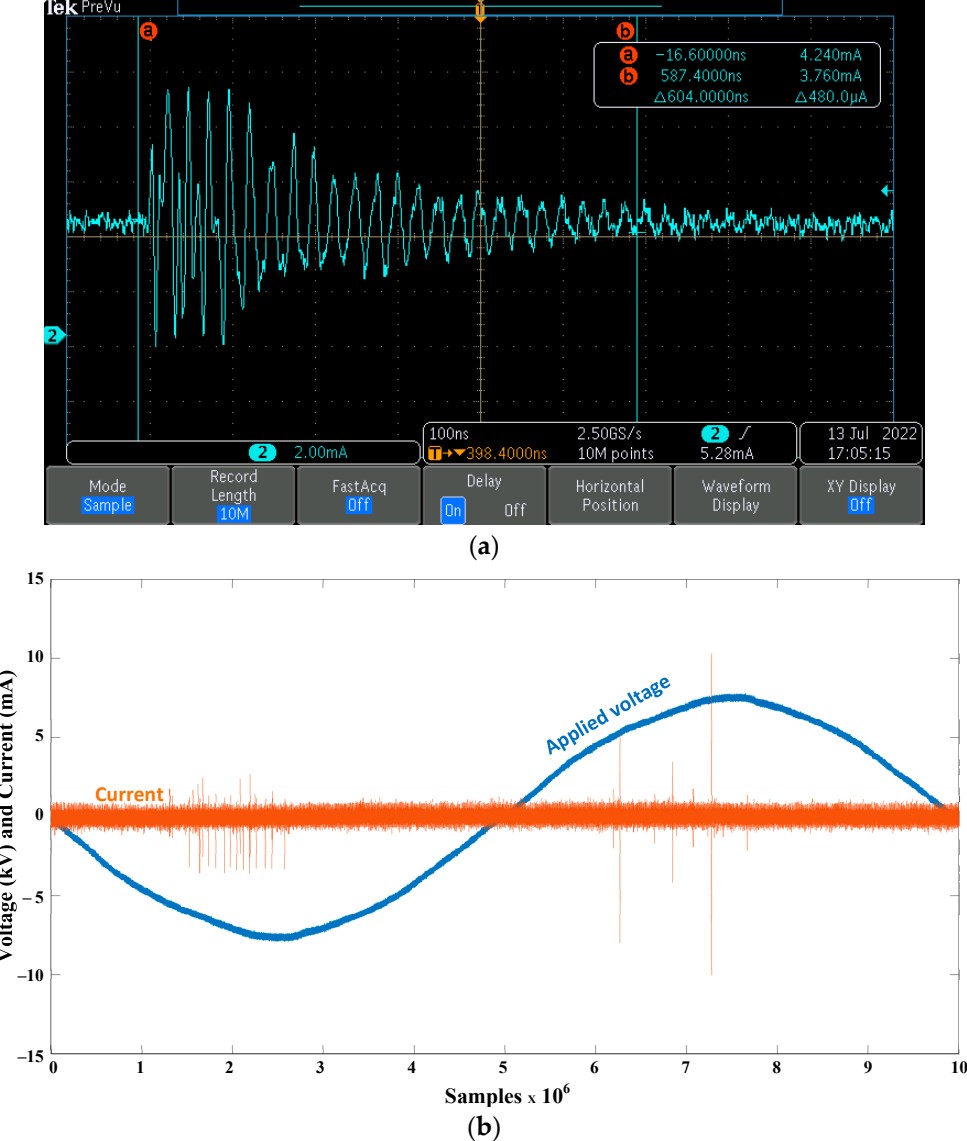

(**a**)

(**b**)

**Figure 1.** (**a**) Current profile of a discharge acquired with an oscilloscope. (**b**) Current and voltage profiles during a period of the voltage waveform acquired at a rate of $10^7$ samples per period of the supply voltage.

To determine the energy of the discharges, the following strategy was applied, which is also summarized in Figure 2:

1. Synchronous acquisition of the voltage and the current over a period of the voltage waveform.
2. Apply a threshold to the acquired current to determine whether or not a discharge occurs and select a window of $N$ points ($N = 200$ points in this work, although it

depends on the resolution of the image sensor and the geometry of the problem) around the threshold value, as shown in Figure 2.

3. Apply the same window to the acquired voltage. All windows must have the same number of points.
4. Calculate the average power of each discharge by applying (1).
5. The electric energy dissipated by the discharges in one cycle of the applied voltage is calculated as:

$$E_{discharges} = \sum_{j=1}^{discharges} P_{discharges}(j) \cdot N \cdot \Delta t \qquad (2)$$

where $P_{discharge}(j)$ is the power of the $j$-th discharge calculated from (1) and $N$ is the number of data points in each considered window (number of points considered in each discharge).

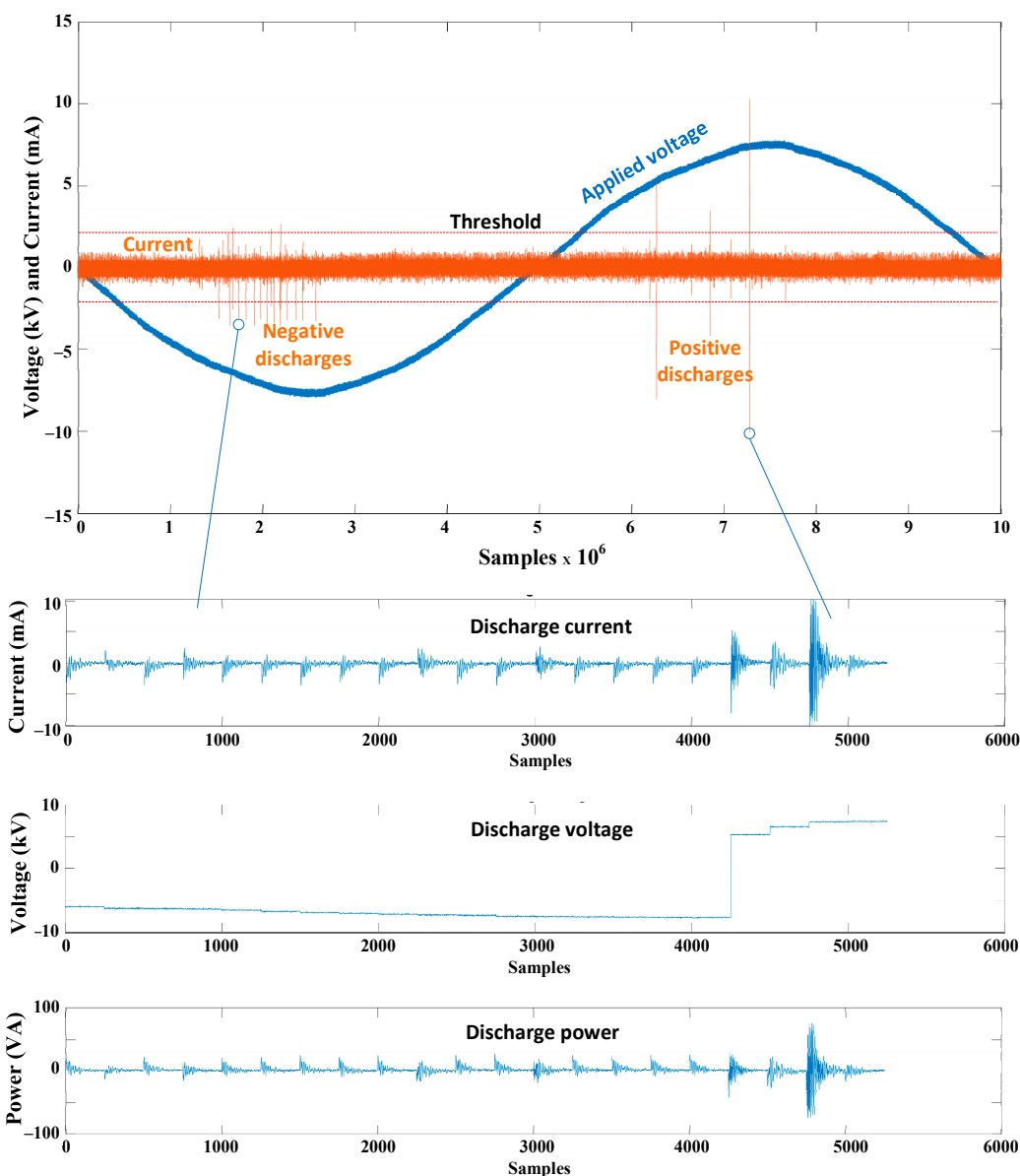

**Figure 2.** Applied data processing strategy to determine the energy dissipated by the discharges.

### 2.2. Intensity of the Digital Images

Current image sensors acquire digital images, $I_{m \times n}$, in the form of a matrix of $m \times n$ pixels. Each pixel is designated as $I(i,j)$, where $(i,j)$ defines the position in the matrix, with $1 \leq i \leq m$, $1 \leq j \leq n$ [33]. The image $I_{m \times n}$ has a resolution of $P = m \times n$ pixels.

All pixels in RGB images contain a triplet (R,G,B) that defines the color coordinates in the RGB colour space, where $R$ = red, $G$ = green, and $B$ = blue, so each pixel is defined by a (1,3) vector. The image results in

$$I_{RGB}(i,j,k) = \{ \underbrace{I_{RGB}(i,j,R)}_{R(i,j)}, \underbrace{I_{RGB}(i,j,G)}_{G(i,j)}, \underbrace{I_{RGB}(i,j,B)}_{B(i,j)} \} \tag{3}$$

where $I_{RGB}(i,j,k)$ is represented by 8-bit unsigned integers, and since $2^8 = 256$, $0 \leq I_{RGB}(i,j,k) \leq 255$.

To determine the intensity of the images, it is common to transform the original RGB image into a grayscale image using the following transformation [34]:

$$I_{grayscale}(i,j) = 0.299 \cdot R(i,j) + 0.587 \cdot G(i,j) + 0.114 \cdot B(i,j) \tag{4}$$

where the weights 0.299, 0.587, and 0.114 of the red, green, and blue channels, respectively, are taken considering the luminous efficacy function of the average human eye, thus approaching the average spectral sensitivity of the human eye.

Once the RGB images have been transformed into grayscale images, their intensity [35,36] can be calculated as:

$$INT_{image}(I_{grayscale}) = \sum_{i=1}^{m} \sum_{j=1}^{n} I_{grayscale}(i,j) \tag{5}$$

The digital image contains sensor and background noise. To reduce the noise effect, a window around the discharge is selected and the effect of the background noise is removed (see Figure 3):

$$INT_{image-background} = \sum_{i=1}^{m*} \sum_{j=1}^{n*} [I_{grayscale}(i,j) - B_{grayscale}(i,j)] \tag{6}$$

where $m^* < m$ and $n^* < n$ are the vertical and horizontal dimensions of the selected window and $B_{grayscale}(i,j)$ is the 8-bit unsigned value of the $i$-th $j$-th position of the background. Equation (6) assumes that in the grayscale image, the value of each pixel (in the 0–255 range) is proportional to the amplitude of the incident light integrated over the exposure time and the pixel area. Since the energy of a wave is proportional to the square of its amplitude, in this paper, it is assumed that the square root of the energy dissipated by the discharges calculated from (2) is proportional to the intensity of the images calculated from (6). Section 5 assesses the accuracy of this approach by means of experimental data.

Figure 3 summarizes the procedure applied to determine the intensity of the discharges calculated from the digital images.

Finally, (6) is normalized to return values within the range 0 to 100,

$$INT_{normalized} = \frac{INT_{image-background}}{n*m*255} 100 = \frac{\sum_{i=1}^{m*} \sum_{j=1}^{n*} [I_{grayscale}(i,j) - B_{grayscale}(i,j)]}{n*m*255} 100 \tag{7}$$

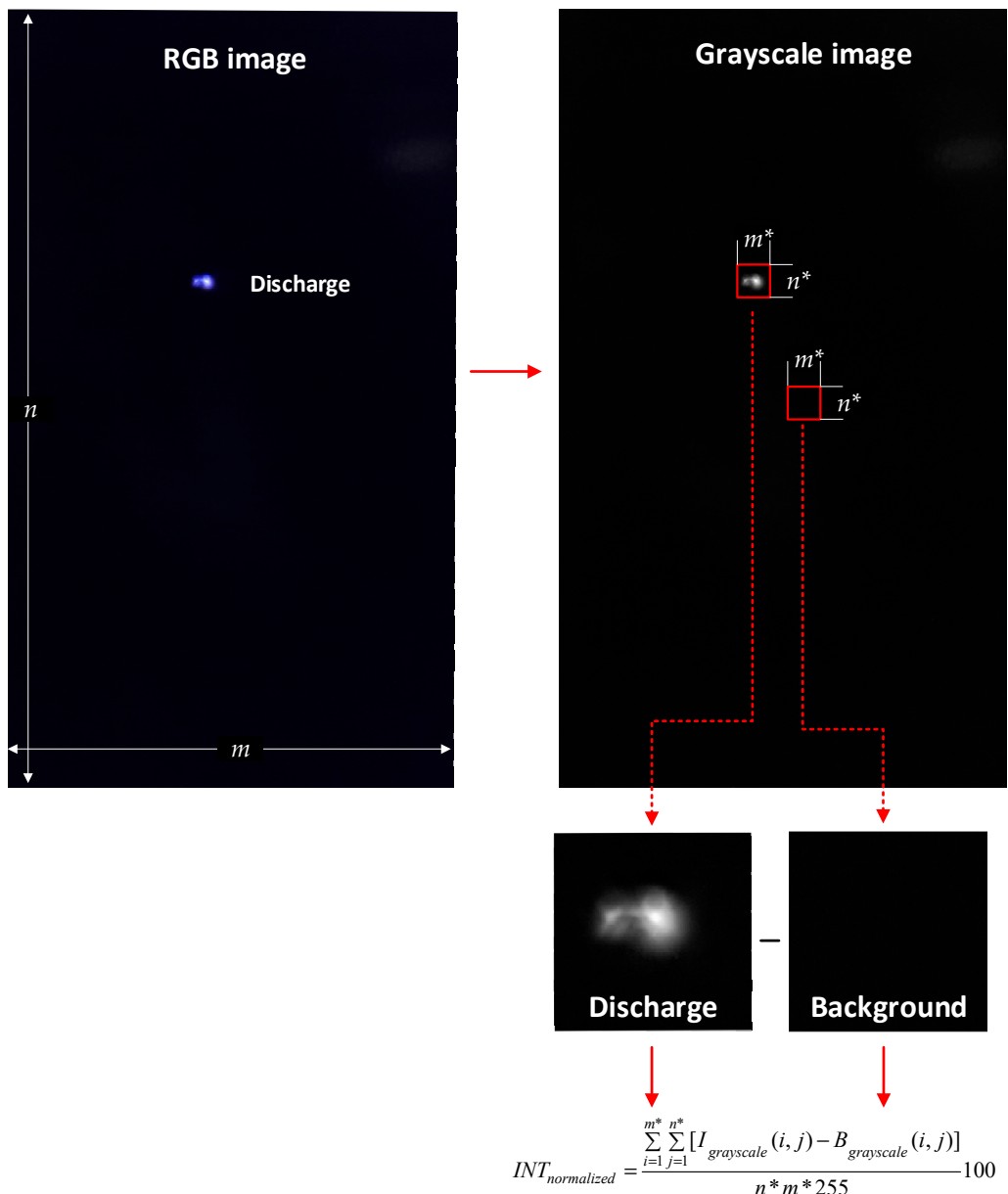

**Figure 3.** Summary of the image processing to determine the intensity of the discharges from digital RGB images.

### 3. Discharge Severity Indicator

The severity of the discharge is related to the intensity or magnitude of the fault [37]. In this paper, it is assumed that the magnitude of the fault can be estimated by using a suitable discharge severity indicator. If it can be verified experimentally that the intensity of the images is directly related to the electric energy dissipated by the discharges, the intensity of the digital images could itself be a simple and fast to calculate severity indicator. Therefore, this paper proposes to use (8) as a discharge severity indicator, which, from (6) and (7), can be expressed as:

$$INT_{normalized} = \frac{\sum\limits_{i=1}^{m*}\sum\limits_{j=1}^{n*}[I_{grayscale}(i,j) - B_{grayscale}(i,j)]}{n*m*255} 100 \tag{8}$$

The adequacy of this approach is assessed in more detail in Section 5.2.

According to (8), the values of the discharge severity indicator $INT_{normalized}$, which are related to the intensity of the discharge, must be within the 0–100 interval, where low values of $INT_{normalized}$ indicate a healthy state without discharges, while high values indicate saturation of the analyzed pixels and thus a high severity of the fault. Section 5.3 presents a detailed analysis of the threshold values to consider.

## 4. Experimental

In this paper, discharges of different intensity or magnitude are generated by increasing the voltage applied to a needle-plane air gap configuration. To emulate pressure conditions in aircraft systems, the discharges are generated inside a low-pressure chamber in the pressure range of 10–100 kPa at 20 °C.

The stainless-steel low-pressure chamber consists of a 375 mm × 260 mm (height × diameter) cylindrical vessel and a methacrylate lid that allows the photographs to be wirelessly transmitted to a computer located near the chamber. The chamber is grounded during the experiments. The low-pressure chamber contains the smartphone used to take the long exposure photographs and the needle-plane gap, as shown in Figure 4.

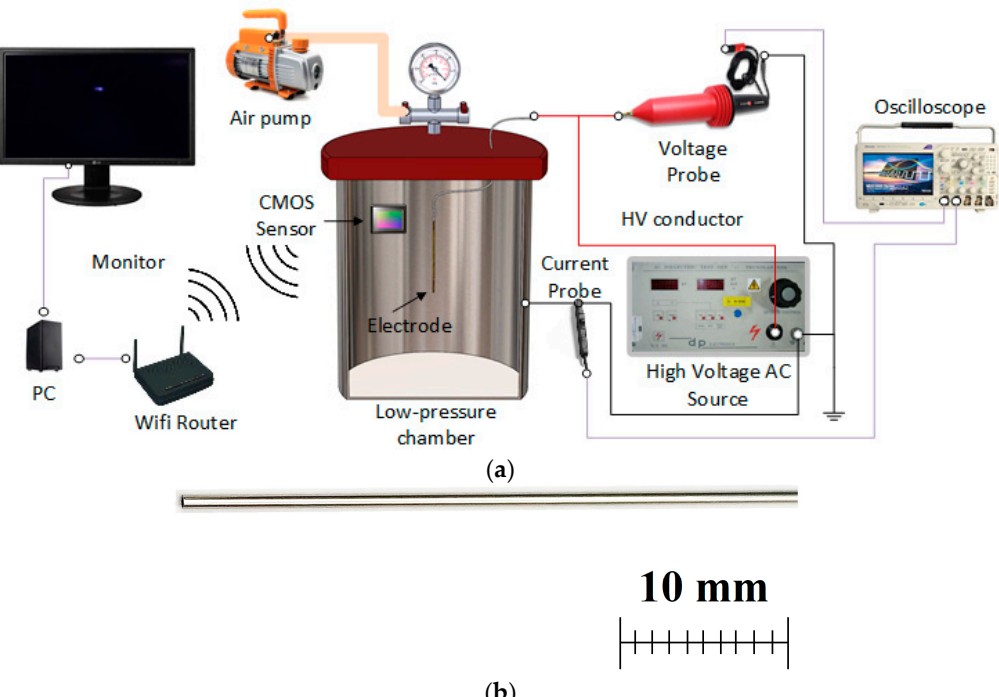

**Figure 4.** (**a**) Sketch of the experimental setup used to reduce the pressure and generate electric discharges. (**b**) Needle-plane air gap (stainless steel, diameter = 0.8 mm).

The pressure inside the chamber was changed using a vacuum pump (BA-1, 1/4 HP, 85 L/min, Bacoeng, Suzhou, China). The ac voltage was changed by means of an adjustable ac high-voltage source (TecnolabRD6, 0–14.4 $kV_{peak}$, 600 VA, dp Electronica SL, Barcelona, Spain).

The applied voltage and leakage current were acquired synchronously using a high-performance oscilloscope (MDO3024, 200 MHz, 2.5 Msamples/s, Tektronix, Beaverton, OR, USA). A high-voltage probe (CT4028, 0–39 $kV_{peak}$, reduction ratio 1000:1, accuracy < 3%, dc to 220 MHz, Electronics, Yorba Linda, CA, USA) was used to adapt the high voltage to the requirements of the oscilloscope. The leakage current circulating through the ground conductor (see Figure 4) was measured using a Hall effect current probe (TCP0030A, dc to >120 MHz, accuracy 1 mA, Tektronix, Beaverton, OR, USA), which was placed around the ground wire to detect the current of the discharges travelling from the high-voltage electrode to the ground terminal.

It is well known that discharges emit near-UV and visible light. In this work, they were acquired by means of a high-resolution back-illuminated CMOS imaging sensor (sensor size 8.0 mm, cell size 0.8 μm × 0.8 μm, 48 Mpixels, lens focal 17.9 mm, raw format images, IMX586, Sony, Tokyo, Japan). This type of sensor was used because it is known to be sensitive to both visible and UV radiation [38,39]. A computer connected to a Wi-Fi router wirelessly controlled the CMOS image sensor, so images could be taken from inside the chamber while it was pressurized. Due to the low intensity of the discharges, to improve the sensitivity of the images, the camera was operated in the long exposure mode (exposure time of 32 s, ISO 800, manual focus).

Corona discharges were generated using a needle-plane air gap, because this geometry is one of the reference gaps used in high-voltage applications [40]. It consists of a 0.8 mm diameter cylindrical stainless-steel needle with a 70 mm long air gap.

## 5. Experimental Results

This section presents and discusses the experimental results obtained in the low-pressure chamber using the instrumentation and air gap geometry described in Section 4.

### 5.1. Digital Images of the Discharges

Long exposure photographs of the discharges (32 s) were taken to increase the number of photons reaching the sensitive elements of the image sensors and thus the sensitivity of the method. Figure 5 summarizes some of the photographs of the discharges acquired under different experimental conditions, that is, under variable pressures within the range of 100–10 kPa and different applied voltages. It is seen that, for a fixed pressure, the intensity of the discharge increases with the applied voltage.

From the photographs shown in Figure 5, it is deduced that the intensity of the images of the discharges increases with the applied voltage, that is, with the electric energy dissipated by the discharges. However, this correlation must be validated using experimental data acquired with the voltage and current probes and by calculating the intensity of the discharges acquired by the CMOS image sensor.

### 5.2. Relationship between the Intensity of the Images and the Electric Energy Dissipated by the Discharges

This section shows that the electric energy dissipated by the discharges is related to the intensity of the digital images. The relationship between the intensity of the images and the electric energy dissipated by the discharges assumed and evaluated in this section is as follows:

$$\underbrace{\sqrt{\overbrace{\sum_{j=1}^{discharges} P_{discharges}(j) \cdot N \cdot \Delta t}}}_{\sqrt{E_{discharges}}} \sim \underbrace{\frac{\sum_{i=1}^{m*}\sum_{j=1}^{n*}\left[I_{grayscale}(i,j) - B_{grayscale}(i,j)\right]}{n*m*255}100}_{INT_{normalized}} \tag{9}$$

The adequacy of (9) is assessed in the next paragraphs. According to (9), the square root of the energy dissipated by the discharges must be proportional to the normalized intensity of the images. It is based on the fact that the amplitude of the signal detected by any individual sensitive element of the CMOS sensor is directly related to the intensity of the light spreading over the sensor. In addition, $INT_{normalized}$ can be used as a discharge severity indicator, providing values from 0 to 100, indicating the degree of severity.

Figure 6a summarizes the experimental relationships between $INT_{normalized}$ and the applied voltage, while Figure 6b shows the experimental measurements obtained with the photometric ($INT_{normalized}$) and electric ($E_{discharges}^{0.5}$) methods and their linear fittings at different operating pressures characteristic of aeronautical environments.

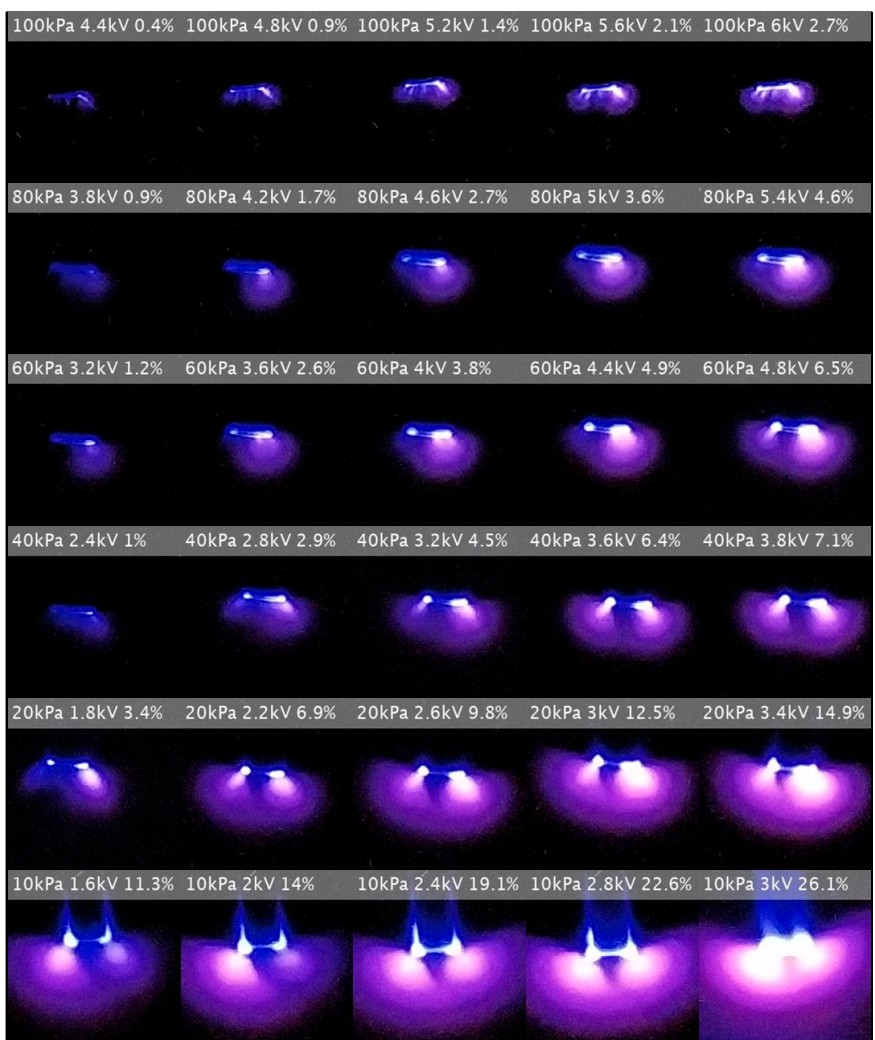

**Figure 5.** Long exposure photographs of the discharges under the experimental conditions described in the images (voltages expressed in RMS value). The images indicate the operating pressure, the applied voltage, and the value of the discharge severity indicator $INT_{normalized}$.

The experimental results presented in Figure 6a show a linear behavior between $INT_{normalized}$ and the applied voltage. Since the intensity of the discharges increases with the applied voltage, these results show that $INT_{normalized}$ is related to the severity of the discharges.

The results presented in Figure 6b show a linear behavior between $INT_{normalized}$ and $E_{discharges}^{0.5}$, thus validating the hypothesis expressed in (9). Table 1 summarizes the parameters of the linear fittings between $INT_{normalized}$ and $E_{discharges}^{0.5}$, where high values of the coefficient of determination $R^2$ can be seen, thus indicating a high correlation between $INT_{normalized}$ and $E_{discharges}^{0.5}$. Therefore, it can be concluded that the electric energy dissipated by the discharges is proportional to the intensity of the digital images.

**Table 1.** Main parameters of the linear regression between $INT_{normalized}$ and $E_{discharges}^{0.5}$ ($INT_{normalized} = INT_o + k \cdot E_{discharge}^{0.5}$).

|  | Parameters | 100 kPa | 80 kPa | 60 kPa | 40 kPa | 20 kPa | 10 kPa |
|---|---|---|---|---|---|---|---|
| **ac supply** | $INT_o$ | −0.3372 | −0.3504 | −0.8642 | 0.1207 | 2.3452 | 7.3545 |
|  | $k$ | 556.52 | 833.43 | 1070.4 | 1156.4 | 1378.8 | 1511.3 |
|  | $R^2$ | 0.9936 | 0.9919 | 0.9982 | 0.9896 | 0.9909 | 0.9640 |

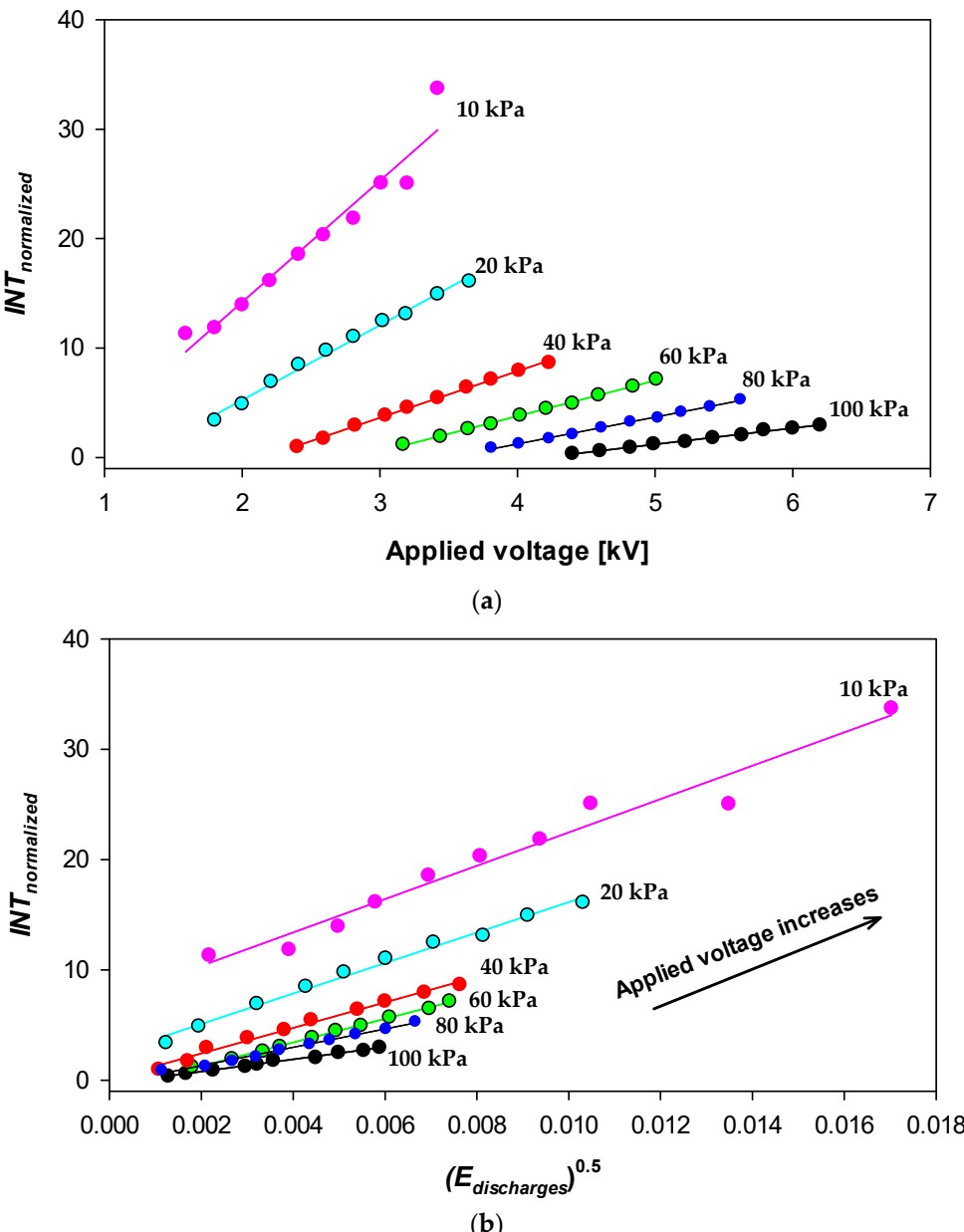

**Figure 6.** Experimental results. (**a**) Correlations between the normalized intensity of the digital images of the discharges and the applied voltage. (**b**) Correlations between the normalized intensity of the digital images of the discharges and the square root of the electric energy dissipated by the discharges.

Results summarized in Figure 6 and Table 1 clearly show the strong linearity between the normalized intensity of the discharge images $INT_{normalized}$ and the square root of the electric energy dissipated by the discharges $E_{discharges}^{0.5}$. These results confirm that $INT_{normalized}$ can be used as an indicator of the discharge severity.

### 5.3. Criterion to Determine the Severity of the Discharges

Results presented in Section 5.2 show that $INT_{normalized}$ can be used as a discharge severity indicator.

Disruptive discharges were then performed to determine the upper limit of the $INT_{normalized}$ indicator. The results obtained indicate that under the experimental conditions of this work, in case of disruptive discharges, the severity indicator $INT_{normalized,upper}$

is always greater than 50, so an upper threshold value of this indicator can be established as $INT_{normalized,upper} = 50$.

Finally, the lower limit of the $INT_{normalized}$ indicator was tested. Without the presence of electric discharges, due to the noise of the CMOS sensor and the background, the value indicated by the $INT_{normalized}$ indicator is not zero. Tests indicate that under our experimental conditions, this value is less than 0.1, so a threshold lower value $INT_{normalized,lower} = 0.1$ of this indicator can be established.

Results obtained manifest that $INT_{normalized}$ can be applied as a discharge severity indicator, as follows:

1.  When ($INT_{normalized} \leq INT_{normalized,lower}$), it corresponds to a healthy condition, without discharges.
2.  When ($INT_{normalized,lower} < INT_{normalized} < INT_{normalized,upper}$), it corresponds to a fault condition, where discharges occur. In this case, a higher value of $INT_{normalized}$ corresponds to a higher discharge severity.
3.  When ($INT_{normalized} \geq INT_{normalized,upper}$), it corresponds to the worst fault condition, that is, a disruptive discharge.

Therefore, a continuous monitoring of the discharges using a digital camera with additional processing to determine the $INT_{normalized}$ discharge severity indicator allows detection of anomalous conditions and generating an alert, which can be very useful to apply predictive maintenance strategies.

## 6. Conclusions

The low-pressure environments, characteristic of aeronautical applications, favor the inception of electrical discharges in electric and electronic circuits placed in unpressurized areas. However, in their early state, due to the low energy level involved, they are undetected by on-board protections until a major failure develops. This paper has developed an optical method to detect, locate, and quantify the severity of the discharges at the earliest stage. The proposed method is based on detecting the near-UV and visible light emitted by the discharges using a high-resolution and low-cost image sensor. A discharge severity indicator based on the intensity of digital images of the discharges has been proposed and tested, demonstrating its accuracy and ability to quantify the magnitude of the discharges in all phases, from the early stage to the development of disruptive discharges. The behavior of the method and fault indicator proposed in this paper has been validated through experimental data acquired in a low-pressure chamber at different pressures in the range of 10–100 kPa, characteristic of aircraft applications, using a needle-plane air gap geometry and a high-resolution low-cost image sensor. Finally, a criterion for quantifying the magnitude of the discharges based on the discharge severity indicator has been proposed, which allows detecting and quantifying of anomalous conditions, which can be very useful to apply predictive maintenance strategies. The results presented in this work allow the fault to be detected at an early stage, long before irreversible damage occurs, so they can complement the on-board electrical protections, which are currently triggered when the fault is well developed. Further work will include testing the approach proposed in this paper in wire bundles to identify and quantify surface discharges and arc tracking activity at a very early stage. Although the method proposed in this paper has been tested in a simulated aircraft environment, a similar approach can be implemented in other applications where electrical discharges are an issue.

**Author Contributions:** Conceptualization, P.B.-C. and J.-R.R.; methodology, J.-R.R.; software, J.-R.R., P.B.-C. and B.O.; validation, P.B.-C., B.O., Y.A.Q. and M.P.; formal analysis, J.-R.R.; investigation, P.B.-C. and J.-R.R.; resources, J.-R.R.; writing—original draft preparation, P.B.-C. and J.-R.R.; writing—review and editing, J.-R.R.; supervision, B.O., Y.A.Q. and M.P. All authors have read and agreed to the published version of the manuscript.

**Funding:** This research was funded by Ministerio de Ciencia e Innovación de España, grant number PID2020-114240RB-I00, and by the Generalitat de Catalunya, grant number 2017 SGR 967.

**Institutional Review Board Statement:** Not applicable.

**Informed Consent Statement:** Not applicable.

**Data Availability Statement:** Not applicable.

**Conflicts of Interest:** The authors declare no conflict of interest.

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
