# Peer review of "Using CMOS Image Sensors to Determine the Intensity of Electrical Discharges for Aircraft Applications"

_applsci, doi:10.3390/app12178595_

Round 1

Reviewer 1 Report

The manuscript presents an interesting topic of the determination of the intensity of electrical discharges for aeronautical applications. The manuscript is mostly well structured and written. The theoretical and pratical aspects and the simulation's analysis and results are well described and discussed. I recommend publishing this work in its current form.

Author Response

Thank you very much for your positive comments.

Reviewer 2 Report

Overall, this quality of this paper is good. However, results analysis can be improved, especially in Figs 5-6. Furthermore, English should be improved before publication. The reviewer finds so many errors such as

1). In abstraction, "The behavior and accuracy of this indicator has was 

in the laboratory using a low-pressure chamber operating in the 10 ‒ 100 kPa pressure....."

2) Partial discharges (PDs) manifest in solid, liquid and gaseous insulation media between two electrodes, and do not entirely bridge the insulation path between the eletrodes.

3) Mathematical expression in Eq. 7 and 8 is not correct.

4) The quality of Fig. 1 and 2 shoud be improved. 

Author Response

Overall, this quality of this paper is good. However, results analysis can be improved, especially in Figs 5-6. Furthermore, English should be improved before publication.

REPLY: Thank you very much for your comment. English grammar and style have been revised and results analysis 6 has been improved.

The reviewer finds so many errors such as

1). In abstraction, "The behavior and accuracy of this indicator has was

REPLY: Thank you very much for your comment. Now it reads: “The behaviour and accuracy of this indicator has been tested in the laboratory using a low-pressure chamber operating in the pressure range of 10 ‒ 100 kPa, which is characteristic of aircraft applications, analysing a needle-plane air gap geometry and using an image sensor”.

in the laboratory using a low-pressure chamber operating in the 10 ‒ 100 kPa pressure....."

REPLY: Thank you very much for your comment. Now it reads: “The behaviour and accuracy of this indicator has been tested in the laboratory using a low-pressure chamber operating in the pressure range of 10 ‒ 100 kPa, which is characteristic of aircraft applications, analysing a needle-plane air gap geometry and using an image sensor”.

2) Partial discharges (PDs) manifest in solid, liquid and gaseous insulation media between two electrodes, and do not entirely bridge the insulation path between the eletrodes.

REPLY: Thank you very much for your comment. Now it reads: “Partial discharges (PDs) occur between two electrodes insulated by a solid, liquid or gaseous insulation when the local electric field strength exceeds the dielectric strength of the insulation material. PDs do not entirely bridge the in-solation path between the electrodes [8]”.

3) Mathematical expression in Eq. 7 and 8 is not correct.

REPLY: Mathematical expressions in Eq. 7 and 8 have been rearranged.

4) The quality of Fig. 1 and 2 shoud be improved.

REPLY:  Thank you very much for your comments. The quality of the figures has been improved.

Reviewer 3 Report

1- abstract and conclusion must be enhanced to describe the novelty of this work.

2- the introduction must be improved by adding the following recent works:

doi: 10.3390/pr9040627

doi: 10.3390/math9182313

doi: 10.3390/math9212770

doi: 10.3390/math9212821

doi: 10.3390/pr10061072 

https://doi.org/10.1109/ACCESS.2021.3052153 

https://doi.org/10.1109/ACCESS.2021.3061529 

3- discussion of results must be enhanced

Author Response

1.-abstract and conclusion must be enhanced to describe the novelty of this work.

REPLY:  Thank you very much for your comment. We have improved the abstract and the Conclusions according to the Referee’s comments. The changes made are highlighted in yellow colour in the revised version of the manuscript.

2- the introduction must be improved by adding the following recent works:

doi: 10.3390/pr9040627

doi: 10.3390/math9182313

doi: 10.3390/math9212770

doi: 10.3390/math9212821

doi: 10.3390/pr10061072

https://doi.org/10.1109/ACCESS.2021.3052153

https://doi.org/10.1109/ACCESS.2021.3061529

REPLY: It seems that there has been a mistake, since we have been deeply reviewing the papers suggested by the Referee and we have noticed that they are not related to the contents of this paper, since most of them are related to methods for identifying model parameters of solar cells.

3- discussion of results must be enhanced

REPLY:  Thank you very much for your comments. We have enhanced the discussion of the results, and the changes made are highlighted in yellow colour in the revised version of the manuscript.

Reviewer 4 Report

To start, the paper is well-organized and flows nicely. The Introduction is elegant in articulating the problem and posited solution; it also frames the various sections of the paper. The presented methods/techniques/theoretical foundations are quite clear, and the experimental section is unambiguous. The Conclusion is sufficiently robust and the cross-domain application is discussed, but a future work discussion is lacking; this would add nicely to the Conclusion.  Each of the sections are well-developed, and the involved literature review is nicely synthesized.  

The point of the authors is crystalline — increased voltage levels segue to higher stress on electric/electronic components and insulation materials. The involved higher electric stress increases the intensity and probability of various types of discharges (e.g., surface, arc tracking, arcing, disruptive discharges, etc.). These discharges have effects, which, among others, include chemical reactions. These chemical alterations can create a conductive pathway in the polymeric insulations, which facilitates the flow of an electric current, thereby elevating the temperature in the area of the pathway and further adversely impacting the insulation.  This cascading affect promotes arcing activity, and this becomes self-reinforcing as the carbonized pathway becomes more ingrained. 

Commercial aircraft have hundred of kilometers of wires and thousands of wiring components, which are mostly hidden inside various conduits/ducts/troughs. These wires/wiring components/electronic circuits, often located in unpressurized areas of the aircraft, are more susceptible to incidents of electrical discharge. Yet, condition-based monitoring and maintenance is difficult given their placement. As these wires/wiring components/electronic circuit paradigms are often situated in areas of high acoustic/electromagnetic noise, the use of conventional sensors has low efficacy. However, optical sensors offer a practical approach for ascertaining State of Health (SoH) and Remaining Useful Life (RUL). These optical sensors can be used to detect corona discharges. Heuristically, as the intensity of the involved images are an indicator of the severity of the discharge, which in turn is related to the severity of the fault, the approach seems prudent. The authors conducted experiments with a low-pressure chamber, and they found that for a “fixed pressure, the intensity of the discharge increases with the applied voltage.” The potential impact is quite significant, as the optical sensors have efficacy at all phases – early stage to the full-blown development of disruptive charges.

There are some typos throughout. On line 33, the word “sense” is incorrectly spelled. On line 70, the word “cruising” should be used. On line 105, the word “can” is incorrectly spelled, etc.

Author Response

REPLY:  Thank you very much for your positive comments. The typos identified by the Reviewer have been solved. In addition, the English grammar and style have been revised in deep. We have also pointed out the future work in the Conclusions section. The changes made are highlighted in yellow colour in the revised version of the manuscript.